# Oral Submucous Fibrosis: Etiological Mechanism, Malignant Transformation, Therapeutic Approaches and Targets

**DOI:** 10.3390/ijms24054992

**Published:** 2023-03-05

**Authors:** Xiaofeng Qin, Yujie Ning, Liming Zhou, Youming Zhu

**Affiliations:** Key Laboratory of Oral Diseases Research of Anhui Province, Department of Dental Implant Center, Stomatologic Hospital and College, Anhui Medical University, Hefei 230032, China

**Keywords:** oral submucous fibrosis, oral squamous cell carcinoma, myofibroblast, malignant transformation, epithelial–mesenchymal transition, natural compounds, miRNAs, lncRNAs

## Abstract

Oral submucosal fibrosis (OSF) is a chronic, progressive and potentially malignant oral disorder with a high regional incidence and malignant rate. With the development of the disease, the normal oral function and social life of patients are seriously affected. This review mainly introduces the various pathogenic factors and mechanisms of OSF, the mechanism of malignant transformation into oral squamous cell carcinoma (OSCC), and the existing treatment methods and new therapeutic targets and drugs. This paper summarizes the key molecules in the pathogenic and malignant mechanism of OSF, the miRNAs and lncRNAs with abnormal changes, and the natural compounds with therapeutic effects, which provides new molecular targets and further research directions for the prevention and treatment of OSF.

## 1. Introduction

OSF is a chronic and insidious oral disease involving multiple parts of the oral cavity and is generally considered to be a disease related to collagen metabolism disorder. Increased collagen formation and decreased collagen degradation lead to the deposition of collagen fibers in oral tissues, which in turn leads to the development of OSF. Patients often go to the doctor because of burning pain of the oral mucosa, accompanied by ulcers, blisters, a diminished sense of taste, a dry mouth, lip and tongue numbness and other symptoms, such as a serious mouth opening restriction, dysphagia and tongue movement disorder. The disease has a certain malignant potential, which belongs to oral potentially malignant disorders (OPMD). At the same time, OSF is closely related to the occurrence of OSCC [1].

## 2. Epidemiology of OSF

The data published by the World Health Organization (WHO) in 2022 reported that the global prevalence of OSF was 4.96%, with a 95% confidence interval of 2.28–8.62 [2]. At present, OSF is mainly found in the Indian Peninsula, including Southeast Asian countries and regions such as India and Pakistan [2], and in China, it is mainly found in Hunan, Hainan and Taiwan [3]. The disease can occur at any age, but is most common in adolescents and adults under the age of 35. Most of the literature mentioned the high incidence of OSF in women, but some literature suggested that the high incidence of OSF was in men aged 20–40 years [4]. Researchers conducted a 10-year follow-up survey and found a significant increase in the number of cases of OSF in eastern India, especially in men [3].

## 3. Risk Factors and Pathogenesis of OSF

### 3.1. Stimulating Factors

Stimulating factors such as eating pepper, smoking and drinking can aggravate the progression of OSF [5]. A study in Taiwan found that alcohol consumption was associated with a higher risk of OSF, increasing the overall risk of malignant transformation of OPMD by 23% [6]. Studies have shown that there are more than 5000 chemicals in tobacco smoke, among which N-nitrosamine is the main cause of its genotoxic effect [7]. N-nitrosamine can directly produce cytotoxic effects on keratinocytes and fibroblasts [8], trigger oxidative stress and inflammation, and activate immune cells, such as macrophages, lymphocytes and B cells [7]. A large number of reactive oxygen species (ROS) are released while smoking and act on intracellular lipid, protein, polypeptide, nucleic acid and other biological macromolecules, resulting in structural and functional changes of proteins and nucleic acid damage [9]. In turn, the accumulation of oxidative damage can cause cell senescence and further accelerate the production of ROS, forming a vicious cycle. The senescence-associated secretory phenotype (SASP) components like interleukin-1 (IL-1), interleukin-6 (IL-6), and growth regulated oncogene α (GRO-α), induce double-strand DNA breaks in keratinocytes and drive genetic instability [10]. Continuous mechanical stimulation to the oral mucosa, causing trauma to the mucosa, and the inflammatory reaction, is initiated locally and a variety of inflammatory mediators are secreted, such as interleukin-1β (IL-1β), IL-6, interleukin-8 (IL-8), tumor necrosis factor-α (TNF-α) and transforming growth factor β (TGF-β); the injured mucosa further atrophy and cause ulcer development due to long-term inflammation [3]. Mucosal ulceration stimulates fibroblast proliferation and activates the coagulation system. Thrombin can induce the activation of TGF-β1 mediated by alphavbeta1 (αvβ1), alphavbeta3 (αvβ3) and alphavbeta5 (αvβ5) integrins, causing the generation of connective tissue growth factors (CTGFs) in buccal mucosal fibroblasts (BMFs) [11,12].

### 3.2. Vitamins and Trace Elements

Individuals who are anemic and deficient in vitamins A, B, C and iron have an increased risk of OSF [13]. Studies have shown that the levels of vitamin A, C and E in the saliva of OSF patients are significantly decreased, while the activities of superoxide dismutase (SOD) and glutathione peroxidase (GPx) are significantly decreased. These changes are positively correlated with the degree of OSF, thus reflecting the increase in oxidative stress with the progression of the OSF [14]. The level of vitamin A in saliva is positively correlated with the level of serum vitamin A, which has the role of stabilizing mucosa, and a deficiency will lead to the loss of mucosecretory cells and epithelial atrophy. Vitamin C is an important free radical scavenger antioxidant, which plays a protective role by alleviating the ROS-induced response, and its level decreases with the increase of collagen synthesis [14].

The interaction between trace elements in saliva and OSF has been confirmed by some studies; the copper level and copper/zinc ratio in the saliva of OSF patients are significantly increased [15]. The levels of zinc and copper in serum were significantly increased in OSF patients [3,13]. Lysyl oxidase (LOX) has a conserved copper binding site, which is a copper-dependent enzyme [16]. High concentrations of copper can increase the activity of LOX, increase collagen synthesis, catalyze the covalent cross-linking of extracellular matrix (ECM) collagen and elastin, and has a certain anti-dissolution ability, which can enhance the hardness and mechanical properties of the ECM [16,17]. The study found that blood samples from patients with OSF were significantly deficient in zinc compared with healthy controls [3]. Matrix metalloproteinases (MMPs) are a class of zinc-dependent proteins and peptide hydrolases [18], which can specifically regulate protein degradation of the ECM, and more importantly, they are related to tumor invasion and metastasis [19]. The tissue inhibitor of matrix metalloproteinases (TIMPs) is a specific inhibitor of MMPs, which jointly maintain ECM homeostasis. The degradation process of the ECM is affected by the increase in TIMPs, which breaks the dynamic balance between TIMPs and MMPs, reduces the degradation of collagen, leads to the deposition of the ECM, and promotes the occurrence and progression of OSF [3].

### 3.3. Immune Factors

Some scholars believe that mucosal fibrosis may be related to allergic reactions caused by exogenous antigen stimulation [20]. Moreover, T lymphocytes, macrophages and mast cells increased significantly in the connective tissue of OSF, and CD4 lymphocytes were dominant [13]. The levels of profibrotic cytokines such as interleukin-1α (IL-1α), IL-1β, IL-6 and TGF-β1 in OSF serum were significantly increased, while the tumor necrosis factor-γ (TNF-γ) was significantly decreased [3]. Antinuclear antibodies (ANA), smooth muscle antibodies (SMA) and gastric parietal cell autoantibodies (GPCA), which are immune-related antibodies, have tested positive in patients with OSF [21]. The expression of TGF-β in OSF tissues was significantly higher than that of normal oral mucosa, in which TGF-β1 was a major growth factor and the most obvious molecule in the process of fibrosis, and was related to the development of almost all fibrosis lesions, including OSF, in which TGF-β2 also played an important role [22,23]. The signaling pathways of TGF-β/Smad2 and Smad4 have been found to be activated in keratinocytes and myofibroblasts in OSF tissues [24]. TGF-β has also been shown to promote fibroblast–myofibroblast differentiation by inducing the contractile phenotype and upregulating α-SMA [25]. In addition, by activating αvβ6-dependent TGF-β1, tissue fibrosis-related genes were upregulated, inducing oral fibroblasts to differentiate into myofibroblasts [23]. Myofibroblasts are α-SMA expressing contractile cells that migrate to the site of injury at the initial stage of inflammation, are responsible for wound healing, and are the major producers of ECM after injury, including fibronectin 1 (FN1) and collagen [26,27]. Myofibroblasts continued to increase from the beginning to the later stages of OSF, suggesting that myofibroblasts could be used as an indicator to evaluate the severity of the OSF [26]. The continued activation of myofibroblasts is associated with the excessive deposition of the ECM and pathological fibrosis; in addition, methods that inhibit myofibroblast activity have been proven to prevent fibrosis, such as of the lung, liver, etc. [25,28].

### 3.4. Genetic Factors

Studies have found that the frequency of HLA-A10, HLA-B7, HLA-DR3, haplotypes A10/DR3, B3/DR3 and A10/B8 in OSF patients increases. The increased frequency of the HLA-B76 phenotype and the increased frequency of the HLA-B51/Cw7 and HLA-B62/Cw7 haplotypes were also associated with OSF susceptibility [3]. Accumulating evidence supports the significance of inherited family history and genetic predisposition in the pathogenesis of OSF [3,24]. Some individuals are more likely to develop OSF due to their genetic polymorphisms in collagen, MMPs, TIMPs and TGF-β1 [5]. Genomic instability was found in 47% to 53% of OSF samples [29]. Studies have indicated that the genotype of type I collagen is associated with the highest risk of OSF, and the genotypes of the low exposure group associated with OSF are CC of collagen type I alpha 1 chain, AA of collagen type I alpha 2 chain, and TT of collagenase-1 [24]. Single nucleotide polymorphisms (SNPs) in the MMP-3 promoter region and the 5A genotype increase the risk of OSF [4]. The CC allele of the TGF-β1 gene on chromosome 19q was associated with OSF risk, and the AA and GG genotypes of the LOX gene on chromosome 5q were the low and high exposure alleles of OSF, respectively. Cystatin C is encoded by the CST3 gene on chromosome 20p, and its high OSF risk allele is AA in both the low and high exposure groups [24]. CST3 or Cystatin C belonging to the cystatin family and decreased production of cysteine protease inhibitors can enhance the collagen degradation of OSF [30]. Cystatin C is an effective inhibitor of lysosomal proteolytic enzymes and cysteine proteases [30]. Higher concentrations of cystatin are directly or indirectly related to the degradation of ECM and lead to the invasion and metastasis of tumor cells [30]. More importantly, it has been demonstrated that Cystatin C is a new type of TGF-β signaling antagonist [30].

### 3.5. Betel Quid Chewing

Betel quid (BQ) is chewed by over 0.6 billion people globally, especially in Asia, where the BQ chews are formulated in a variety of formulas, but usually include betel nuts, betel leaves and hydrated lime, and often contain tobacco [31]. The International Agency for Research on Cancer (IARC) has reported that BQ is carcinogenic to humans. This has also been confirmed in animal studies, where BQ intake has been linked to OSF and OSCC, as well as other cancers. The chemical composition of BQ has been repeatedly studied and arecoline has been consistently detected in all products; regardless of the type of areca nut product, regional location and maturity of the BQ chewers, the arecoline has been considered a key driver of OSF and OSCC [31]. It has been reported that the relative risk of OSF increases significantly with the increase of BQ chewing frequency, and any frequency of BQ intake will increase the risk of OSF by about 50 times [5].

The crude fiber, areca alkaloids, slaked lime, ROS, copper and other components of BQ play different degrees of promoting role in the process of OSF. Long-term chewing of BQ is both a physical and chemical stimulus. Arecoline is a primary areca alkaloid, which itself induces ectopic changes in homologous chromosome alleles [24], and also leads to chromosome breakage and other cell malformations [32]. Areca alkaloids will also form N-nitrosamine metabolites after nitrosation [3]. Shajedul Islam et al. found that under the joint action of arecaidine and slaked lime, the proliferation of BMFs increased and the phenotype of cells changed, leading to the increase of collagen fiber production and the promotion of collagen synthesis [3,33]. The profibrotic effect of arecoline on BMFs may also be indirectly induced by oral keratinocytes, thereby affecting the collagen metabolism of BMFs [31].

Arecoline activated YAP by increasing the level of ROS and inducing the PERK pathway, leads to the start of the endothelial–mesenchymal transition (EMT) and then OSF [34]. The ROS is generated by the autooxidation of areca polyphenols in the mouth and the nitrification of areca alkaloids [31]. TGF-β1 activation, Smad2 phosphorylation and ROS production in BMFs are induced by arecoline [23]. Previous evidence has shown that CD147 is related to the development of multiorgan fibrosis. On this basis, Wang et al. showed that arecoline promoted the expression of CD147 in human oral keratinocytes through the TGF-β1 signaling pathway, and upregulation of CD147 may promote the development of OSF [35]. Arecoline-induced mitochondrial ROS leads to the initiation of TGF-β1 signaling in human oral mucosal fibroblasts, and subsequently, increases the composition of CTGF and Egr1, which are key TGF-β factors in fibrotic diseases [23]. Arecoline can directly stimulate BMFs to synthesize collagen and differentiate into myofibroblasts [23].

More soluble copper can be detected in the oral environment of QB chewers and can be absorbed by buccal mucosa, which will promote the process of OSF [36]. Plasminogen activator inhibitor Type-1 (PAI-1) inhibits MMPs activation to adjust the dynamic balance of the ECM, and excessive PAI-1 can aggravate fibrosis [37]. TGF-β may promote the expression of PAI-1 through ROS and Smad-dependent (ALK5/Smad2/3) and Smad-independent (Src/EGFR/MEK/ERK) pathways [24]. Hypoxia increases arecoline-induced production of PAI-1 and ECM in oral mucosal fibroblasts [24]. Hypoxia-induced factor-1α (HIF-1α) promotes epithelial–mesenchymal transition by increasing extracellular matrix modifiers and LOX, leading to OSF fiber formation [3]. ROS and HIF-1α can induce upregulation of TGF-β1 under hypoxia conditions [3].

### 3.6. Specific RNAs to OSF

In the past two decades, non-coding RNAs (ncRNAs), which do not encode proteins, have been found to play a key role in physiological and pathological processes by regulating gene transcription and translation through various mechanisms, such as microRNAs (miRNAs), long non-coding RNAs (lncRNAs), circular RNAs (circRNAs), etc. [38].

Yang et al. observed that the expression of miR-29c in fibrotic buccal mucosa fibroblasts (fBMFs) was downregulated and that transfection of miR-29c mimics reduced the fBMFs migration ability and collagen gel contractility whereas inhibition of miR-29c could induce a myofibroblast phenotype, and mir-29c also inhibited the activation of myofibroblasts by inhibiting tropomyosin-1 (TPM1) [28]. From the above results, it is not difficult to see that upregulation of miR-29c can delay the development of OSF, so miR-29c can be regarded as a promising therapeutic target for OSF. MiR-21 is a definite non-coding RNA for fibrosis. Liao et al. verified that programmed cell death factor 4 (PDCD4) is an immediate target of miR-21, and the overexpression of PDCD4 in fBMFs attenuates the activity of myofibroblasts [39]. The study by Han et al. clarified that adipose-derived stem cell-derived extracellular vesicles (ADSC-EVs) inhibited the proliferation, migration, invasion and fibrosis of fBMFs and promoted apoptosis through the miR-375/FOXF1 axis, thus inhibiting OSF progression [40]. Chattopadhyay et al. showed that the expressions of miR-31 and miR-204 were, respectively, upregulated and downregulated in OSF tissues [41]. Chickooree et al. summarized the expression profiles of miRNAs in the buccal mucosa of clinically significant OSF patients and normal volunteers and found that there existed clear differences in the expression of 11 miRNAs. Among them, the overexpressed miRNAs were hsa-miR-455-3p, hsa-miR-455-5p and hsa-miR-623, and the underexpressed miRNAs were hsa-miR-1290, hsa-miR-3180-3p, hsa-miR-4792, hsa-miR-509-3-5p, hsa-miR-5189, hsa-miR-610, hsa-miR-760 and hsa-miR-921 [42].

In the above papers, some scholars have verified the targets of miRNAs (miR-760 [42,43], miR-455 [42,44], miR-29c [28], miR-21 [39], miR-10b [45], miR-200C [46], miR-1246 [47], miR-203 [48]). However, there are still some miRNAs (miR-760 [42,43], miR-455 [42,44], miR-509-5p [42], miR-610 [42], miR-10b [45], miR-623 [42], miR-31 [41], miR-204 [41]) whose target sites are only predicted by TargetScan, miRanda and other databases. However, the exact targets of these miRNAs, the signaling pathways involved in them, and whether they have the effect of inhibiting the proliferation and migration of OSF cells, promoting the apoptosis of the OSF cells as predicted, need to be verified by subsequent experiments.

Lee et al. showed that LINC00084, as a sponge of miR-204, upregulated the expression of zinc finger E-box binding homeobox 1 (ZEB1) and induced the transdifferentiation of myofibroblasts, causing an increase of ECM and fibrosis [49]. Therefore, downregulation of LINC00084 expression is considered to reduce the continuous activation of fBMFs and further prevent the malignant transformation of OSF. Yu et al. showed that the chronic stimulation of areca nut activates TGF-β signaling, leading to upregulation of the lncRNA H19 expression, thereby preventing the type I collagen inhibition of Mir-29b and reducing the activity of myofibroblasts [50]. Therefore, targeting the molecules on the H19/miR-29b axis is a possible way to alleviate OSF. Zhou et al. found that exosome-derived lncRNA ADAMTS9-AS2 inhibited the progression of OSF through the AKT signaling pathway, and also inhibited the PI3KT-Akt signaling pathway and EMT [51].

According to the above, we proposed a multi-factor pathogenic model of OSF with BQ as the main cause (Figure 1). The above key factors involved in the occurrence and development of OSF can become therapeutic targets for the disease and block the development of the disease. However, because TGF-β, HIF-1α and other molecules involve too many pathways, their regulatory effects on the human body are too extensive. Therefore, relatively small molecules, such as LOX, MMPs, TIMPs, Cystatin C, PAI-1, CTGF, Egr-1 and ncRNAs, can be considered for further research, and can improve local inflammation, the hypoxia environment and oxidative stress, so that they can be applied in clinical practice quickly.

## 4. Malignant Transformation of OSF

Oral cancer refers to malignant tumors of the mouth and lips, about 90% of which are diagnosed as OSCC [1]. In the GLOBOCAN 2020 report, there were 377,713 new cases of oral cancer (2% of the total 36 cancer cases) and 177,757 new deaths (1.8% of the total 36 cancer deaths) [52]. The incidence is highest in Melanesia and Central and South Asia, among which 69.1% are male, and there is a large regional difference [52]. Patients with OSF can have accompanying oral leukoplakia, oral lichen planus and other OPMD. OPMD lesions are more prone to cancer due to field cancerization. Therefore, OPMD is an important group of mucosal diseases before the diagnosis of OSCC [53]. Abnormal epithelial growth and epithelial atrophy can increase the probability of carcinogenesis, and current studies have shown a relevance between OSF epithelial dysplasia and malignant transformation [54]. Other studies have shown that the thickness of fibrosis in OSF is correlated with epithelial dysplasia [55]. The WHO reports that oral cancer is more common in areas of the world where BQ is chewed. South Asia has a large number of patients with OSF who develop oral cancer, with a cancer rate of about 4.2% (CI 2.7–5.6%) [1]. Tobacco, wine and betel nut are the main causes of oral cancer [56]. The carcinogenic component of areca nut can also induce gene mutations, such as tumor suppressor gene inactivation and activation of oncogenes, which can cause the development of cancer. Tobacco is a known carcinogen, and previous studies have shown that nitrosamines found in tobacco products are the main cause of their genotoxic effects [42]. Tobacco is metabolically activated by P450 enzymes to form N-nitronicotinoids, which can cause DNA damage and lead to potentially malignant diseases and, ultimately, oral cancer [7]. In patients with OSF, the progression to OSCC is associated with smoking and alcohol, but specific mechanisms of fibrosis are also involved [5]. Some studies have pointed out that OSCC originating from OSF tends to be younger and more aggressive [3].

## 5. Mechanism of OSF Malignant Transformation into OSCC

BQ and its additive tobacco consumed by OSF patients are both first-level carcinogens that have been clearly defined by the IARC, which can damage cell genes, cause radiation damage, impair immune function, and further induce OSCC [57,58]. The potential carcinogenic components of BQ are mainly alkaloids and polyphenols. Although polyphenols have antioxidant effects, they may enhance the genotoxic and carcinogenic effects of alkaloids under certain environmental conditions, which is of special significance for the development of OSCC [31]. Arecoline is proved in OSCC and OSF-induced inflammation and produces ROS [31]. Slaked lime works by changing the environmental conditions, changing the pH of polyphenols, allowing them to be oxidized to produce ROS, regulating cell proliferation, cell migration and invasion, and potentially promoting cancer. And more importantly, nitrosamines interact with other macromolecules of the cell through oxidative stress, thereby promoting the occurrence of oral cancer [31,33]. The above results emphasize the importance of BQ components, including tobacco, in the malignant transformation of OSF.

ROS are a group of short-lived, highly active, oxygen-containing molecules that can induce DNA damage and influence DNA damage response [59]. Numerous previous studies have shown that a small amount of ROS can adjust intracellular signal transduction and maintain a dynamic balance. However, a large number of ROS play a key role in the destruction of proteins and DNA, and even induce the development of cancer [9]. During the progression of OSF, a large number of ROS are produced, including both exogenous (BQ and tobacco) and endogenous ROS. The imbalance between ROS and the antioxidant defense system will cause oxidative stress, thus initiating the occurrence of cancer, and the ROS is joined in the adjustment of cancer cell apoptosis [60]. More importantly, studies have shown that in any normal cell, high levels of ROS can transform it into malignant cells [60]. A recent article by Nithiyanantham et al. showed that arecoline N-oxide plays an initial carcinogenic role in the oral cavity through inflammation, consumption of ROS and antioxidant enzymes [61]. ROS can regulate a lot of signaling pathways through transcription factors, such as nuclear factor-kappa-B (NF-κB), signal transducer and activator of transcription 3 (STAT3), HIF-1α, kinases, growth factors, cytokines and other proteins and enzymes, which are related to cell transformation, inflammation, tumor survival, proliferation, invasion, angiogenesis and tumor metastasis [60]. The ROS is associated with epigenetic changes in genes, which can help diagnose diseases [60]. The dual role of the ROS in tumorigenesis and progression includes ROS-dependent malignant transformation and oxidative stress-induced cell death [62], which provides us with new ideas for the prevention and treatment of OSF and OSCC. The research on ROS as a therapeutic target is worthy of further study. The use of antioxidants in the early stage can inhibit the ROS and prevent the occurrence of OSF and the activation of the OSCC tumor signaling pathway. After the disease, ROS production can be promoted, so that it can play the role of oxidative stress against the cancer cells to induce cancer cell death.

Senescence is the common feature of wound healing, fibrosis and cancer [10]. Although senescence has a temporary anti-fibrosis effect, prolonging the senescence will promote fibrosis and malignant transformation [10]. Fibroblast senescence is induced by ROS production by keratinocytes in a TGF-β-dependent manner [31]. Previous studies have demonstrated that senescent fibroblasts have the same characteristics as activated fibroblasts/myofibroblasts and could promote the progression of OSCC through the production of ROS and MMPs [31]. The SASP derived from myofibroblasts induces the EMT of OSF and promotes cancer progression [10]. The senescence of oral mucosa cells plays an irreplaceable role in promoting the malignant transformation of OSF, and it also interacts with other cancer-related factors in OSCC, playing a synergistic role.

EMT is a cellular process that transforms epithelial cells into a mesenchymal phenotype and is critical for tumor migration, tumor stem cell properties, chemotherapy resistance and metastatic potential [63]. During the dynamic transformation of EMT, E-cadherin, occludins, α-catenin and claudins, which maintain epithelial cell–cell junctions, are downregulated, whereas mesenchymal markers are overexpressed, such as α-SMA, snail, slug, vimentin, MMPs, insulin-like growth factors 1 (IGF-1), ferroptosis-suppressor-protein 1 (FSP-1), N-cadherin and zinc finger E-box binding homeobox (ZEB) [64,65,66]. These changes resulted in the loss of cell adhesion, increased ECM composition, enhanced migration potential and increased invasiveness [64]. Existing research also demonstrated that OSF significantly increased the cell–matrix adhesion, invasion and migration abilities and the activity of the MMP2 and IGF-1R of oral cancer, and affected the EMT by enhancing the expression of N-cadherin, fibronectin and vimentin and downregulating the expression of E-cadherin in human oral cancer cells [67]. Researchers have suggested that turning off the expression of the EMT-induced transcription factor Twist1 to reverse EMT is vital for spreading tumor cell proliferation and promoting distant colonization [68]. EMT activation plays an important role in the initial metastasis of OSCC and generates cancer stem cells (CSCs) in OSCC [69]. CSCs, also known as cancer initiating cells, have the ability to self-renew and are a risky cancer cell population associated with cancer disease recurrence, aggressiveness and resistance to chemoradiotherapy [70,71]. At this point, it has to be said that to improve the cure rate of OSCC, reduce the recurrence rate and mortality, targeting the CSCs is crucial.

Several articles demonstrated that arecoline activates the TGF-β pathway [22], and the TGF-β signal transduction pathway is a key pathway to trigger OSCC [12]. The phosphatase and tensin homolog (PTEN) plays a role in immunity, fibrosis, malignancy and is inversely correlated with α-SMA [72]. The increase of TGF-β in OSF can reduce the level of the PTEN, and the inactivation of the PTEN gene leads to the upregulation of the AKT/S6K/Snail pathway by TGF-β1, resulting in the disassembly of the tight junctions of epithelial cells, the destruction of the basement membrane and the increase of epithelial-derived myofibroblasts, which play a role in EMT [72]. At the same time, AKT activity was enhanced, which prolonged the survival time of fibroblasts, and increased ECM production and fibrosis [72]. Richter et al. found that long-term combined stimulation of TGF-β1/EGF could enhance the invasive phenotype of OSCC compared with single growth factor stimulation, such as the significantly upregulated expression of MMP2 and MMP9 [73]. Increased expression of CD105, the TGF-β1 receptor, is associated with hypoxia-induced angiogenic activity in OSF and with the transformation of epithelial dysplasia [74,75,76]. One of the predictive methods for the malignant transformation of OSF is the presence of epithelial dysplasia on the initial biopsy [77].

Another factor in the malignant transformation of OSF is the HIF-1α. At this stage of OSF, collagen fiber accumulation, degeneration and vascular occlusion in the lamina propria and submucosa make oxygen supply to the diseased tissue more insufficient. HIF-1α, a tumor growth signal that is upregulated during hypoxia, is responsible for activating the vascular endothelial growth factor pro-angiogenic gene, which has an effect on tumor growth, glucose metabolism, invasion, chemoradiotherapy resistance and prognosis [78].

Studies have also confirmed that hypoxia modulates the activity of three key transcription factors (c-Myc, p53 and HIF-1α), leading to the continuous accumulation of ROS and the malignant transformation of cells [60]. Ishida et al. showed that hypoxia-induced EMT in OSCC occurs through the activation of the Notch signaling pathway [79]. Therefore, it is necessary to reduce the expression of HIF-1α in OSF and OSCC by targeting HIF-1α as a target, both in prevention and treatment.

The inflammatory environment and degree of fibrosis also play a contributing role in the malignant transformation of OSF. Microtrauma in the process of OSF accelerates the diffusion of chemical components in the BQ into the submucosal tissues, and immune cells are locally recruited and secrete proinflammatory cytokines [80], resulting in inflammatory cell infiltration in submucosal tissues, which causes further atrophy and ulcers of mucosa, and continuous tissue inflammation leads to tissue fibrosis and malignant transformation. Various articles have told us that increased myofibroblasts are related to the severity and progression of OSF and OSCC [49]. Bale et al. have also demonstrated that the degree of fibrosis and levels of oxidative stress biomarkers (serum malondialdehyde (MDA) and SOD) are associated with malignant transformation [81].

The study by Reis et al. showed that the loss of PDCD4 expression was associated with tumorigenesis and invasion of OSCC, and the downregulation of PDCD4 by miR-21 was proved to increase the invasivity of oral cancer [82]. As a target gene of miR-31, the downregulation of C-X-C Motif Chemokine Ligand 12 (CXCL12) was of great significance in the development of precancerous lesions to cancer [41]. Wang et al. found that the circEPSTI1/miR-942-5p/LTBP2 axis promoted the proliferation and invasion of OSCC cells and promoted the progression of OSCC via the phosphorylation of the PI3K/Akt/mTOR signaling pathway elements [83]. Studies have shown that lncRNA ADAMTS9-AS2 inhibits PI3K-AKT signaling and EMT in OSCC. Exosomal ADAMTS9-AS2 can be transported into cells and exerts a similar tumor suppressor effect as exogenous ADAMTS9-AS2, which provides further evidence that exosomal lncRNAs can have a vital role in the occurrence and development of OSCC [51]. Chen et al. showed that IncRNA MEG3 inhibits the self-renewal and invasion ability of oral cancer stem cells by interacting with the molecular sponge miR-421, and the low expression of MEG3 can serve as a marker of poor prognosis in oral cancer [84].

Existing evidence has long been clear that OSF can go through malignant transformation into OSCC, and OSCC of OSF origin is clinically more aggressive than OSCC of non-OSF origin, with a higher rate of metastasis and recurrence [32]. OSF is an irreversible disease, even if the pathogenic factors are removed, so it is not only necessary to block the occurrence of OSCC from the malignant transformation process of OSF, but more importantly, to prevent OSCC from the occurrence of OSF. Figure 2 is our proposed OSCC model of malignant transformation based on OSF.

## 6. Diagnosis

### 6.1. Clinical Diagnosis

Patients generally have a history of chewing BQ, and the clinical symptoms are white oral mucosa accompanied by leather-like texture changes. Patients will have severe burning in the oral cavity after eating spicy food [85]. With the development of this illness, patients may have dysfunctions in eating, chewing, pronunciation and even other functions in the latter course of the disease, which seriously affect patients’ nutrition intake and social communication.

### 6.2. Histological Diagnosis

The main pathological changes in OSF include epithelial atrophy, collagen fiber accumulation, degeneration, vascular occlusion and reduction in the lamina propria and submucosa of the mucosa, and some patients experience epithelial dysplasia [23]. In patients with severely impaired mouth opening, numerous muscle fiber necrosis can be seen. Electron microscopic examination showed that the space between the epithelial cells was widened, a lot of free desmosomes or cell debris could be seen, the number of mitochondria was reduced, some mitochondria were swollen, and collagen fibers with hyalinosis were distributed in bundles [2].

### 6.3. Other Diagnostic Methods

Tissue biopsy is the gold standard, but it is invasive. So, we need to find easier, less invasive, more accurate and less expensive screening and diagnosis methods. Some biochemical parameters are changed in the process of OSF, which suggests that these biomarkers can be used as tools for disease progression and detection of malignant transformation.

Shaikh et al. used attenuated total reflection–Fourier transform infrared spectroscopy (ATR-FTIR) combined with salivary total protein measurement to distinguish between OSF patients and healthy controls [86]. Analysis by Singh et al. showed that people with blood type A were more likely to develop oral cancer and OPMD, while people with blood type O were less likely to develop oral cancer [87]. Therefore, blood typing could be used to identify susceptible people for early disease prevention and surveillance. The clinical analysis of OSF patients by Bale et al. concluded that the serum MDA level increased with the increase in clinical stage, while the serum SOD level decreased with the increase in clinical stage [81]. The above two measurements can be used to evaluate the level of oxidative damage caused by this illness, and also could serve as a diagnostic method to prevent malignant transformation of OSF [81]. Other known markers of fibrosis include TGF-β [88], IL-6 [89], transglutaminase 2 (TGM2) [74], α-SMA [75], collagen type I and collagen type III, etc. Kamala et al. showed that the expression of the Ki-67 antigen increased from normal oral mucosa to OSF and OSCC. In addition, the expression of Ki-67 increased with the aggravation of dysplasia [77]. The results from Nag et al. said that the expression of p63 and E-cadherin and the number of mitotic figures could be used as molecular markers to evaluate the malignant potential of OSF [76]. Of course, not only these, but also p53 [90], cyclin D1, β-catenin, Rb protein, B-cell lymphoma-2 (Bcl-2), the Bcl-2 associated X protein (Bax), the cellular mesenchymal epithelial transition factor (c-Met) and PTEN are diagnostic markers that can be used to predict the malignant transition of OSF [91]. Monteiro et al. also used the combined biomarker to predict high-risk OSF, and proposed a combined expression level formula: 0.688 × Ki-67 + 0.888 × p16 [72]. This provides a new idea for the application of OSF biomarkers. Of course, both the current biomarkers and formulas, and the additional indicators that will emerge later, need to be tested and verified with a large number of samples before they can be used in clinical practice.

## 7. Disease Management

OSF is an irreversible disease, even if the pathogenic factors are removed in the latter stage, so early treatment will have a better prognosis and prevent malignant transformation as much as possible.

### 7.1. Oral Health Education

The incidence of OSF is closely related to chewing BQ. Health education should be strengthened to enhance people’s understanding of the potential harm from chewing BQ. Patients with clinical symptoms should be treated in stomatological hospitals as soon as possible.

### 7.2. Removal of Pathogenic Factors

Quit the chewing BQ habit, quit smoking, quit alcohol and avoid spicy food stimulation.

### 7.3. Drug Therapy

The therapeutic principles of OSF mainly include anti-inflammatory, anti-fibrosis, improvement of ischemic state and anti-oxidation. Commonly used OSF therapeutic drugs mainly include the following categories:(1)Glucocorticoids: glucocorticoids can inhibit the production of inflammatory factors and promote the apoptosis of inflammatory cells, thereby playing the role of anti-inflammatory and inhibiting the process of fibrosis [92,93];(2)Antifibrotic drugs and proteolytic enzymes: exogenous antifibrotic factors and proteolytic enzymes can reverse the process of OSF fibrosis [93]. In clinical practice, hyaluronidase is often used in combination with hormones, and clinical studies have shown that dexamethasone combined with hyaluronidase is the clinical efficacy of local injection into lesions [93];(3)Peripheral vascular dilators: improve the microcirculation and hemorheology in the lesion area to improve clinical efficacy and relieve the symptoms of patients with OSF [94,95]. Clinical studies have shown that treatment with oral isoxsuprine and combined injections of dexamethasone and hyaluronidase are more effective in relieving OSF symptoms than treatment alone [95];(4)Antioxidants and nutritional elements: in the treatment of OSF, the use of antioxidants and nutritional elements can reduce the damage to macromolecules caused by reactive oxygen species, thereby slowing down the progression of OSF [96,97].

### 7.4. Auxiliary Mouth Opening Training

With the help of oral physiotherapy exercise, tissue elasticity and mouth opening can be increased [98]. There are researchers who verify the effect of mouth opening training, the control group using salvia miltiorrhiza combined with triamcinolone local injection treatment and the experimental group, on the basis of the control group, combined mouth opening training for 2 years. The oral opening of the two groups at the end of local injection treatment, 1-year and 2 years after treatment, was compared. The oral opening of the experimental group was significantly higher than that of the control group, and the effective rate of the experimental group was 97.1%, which was significantly higher than that of the control group at the return visit 2 years later [99].

### 7.5. Surgical Treatment

The first choice is surgical treatment for advanced patients, who have limited mouth opening movement due to heavy fibrous bands in the mucosa. The operation included excision of the fibrous band and insertion of different flaps, such as a palatal flap, tongue flap, nasolabial flap, platysma myocutaneous flap, etc. [92]. Platysma musculocutaneous flaps have good flexibility and the same texture as oral mucosa. However, their high technical sensitivity and postoperative complications, such as flap necrosis, flap cracking, skin paddle loss and donor area lesions, are more common [100]. A split skin graft is widely used as a graft material, but postoperative wound contracture and scarring can lead to recurrence of submucosal fibrosis [101]. The buccal fat pad is easy to obtain and can be used to cover the defective area after the removal of the fibrous band. However, the tissue size of the flap is limited, which may not be enough to cover the defect completely, and there may be mild secondary fibrosis in the later stage [102]. The blood supply of the nasolabial flap was sufficient and the defect adaptability was good, but scar and hair growth outside the mouth could be observed [102]. Different skin flaps should be selected according to the specific conditions of the patients with the removal of the fibrous bands. Clinical combined reconstruction has been used to repair the defect, such as the joint reconstruction of the buccal fat pad and nasolabial flap [101]. Of course, postoperative mouth opening training with drug therapy is also needed to improve the prognosis.

### 7.6. Hyperbaric Oxygen Therapy, Laser Therapy, Natural Compounds

In addition to the above conventional treatment methods, in recent years, some scholars have adopted new treatment methods for the clinical manifestations and pathogenesis of OSF, such as hyperbaric oxygen therapy [103], laser therapy [104] and new functions of some natural compounds.

Hyperbaric oxygen can increase blood oxygen content, improve local ischemia and hypoxia, promote neovascularization and collateral circulation in the affected area [103]. Existing clinical studies have shown that laser therapy can alleviate the difficulty in opening the mouth, the burning sensation and even increase cheek flexibility in patients with OSF [104].

Arctigenin is a lignan extracted from Arctium lappa, which has a lot of pharmacological effects, including antifibrotic effects [12]. The results of Lin et al. suggest that arctigenin could inhibit arecoline stimulated TGF-β/Smad2 signaling and reduce fBMFs activity [12]. In addition, burdockaglyin has been proven to reduce the expression of LINC00974, which could activate the TGF-β/Smad signaling pathway to promote the occurrence of oral fibrogenesis [12]. At present, previous studies have shown that salvia miltiorrhiza can improve blood supply, inhibit collagen accumulation, the proliferation of fibroblasts and EMT, and has therapeutic effects on OSF [105]. Black turmeric is commonly known Kali Haldi. Moreover, Bohra et al. found that Kali Haldi and Aloe vera have a synergistic effect on antioxidant radicals in patients with OSF, and they are even better at correcting the burning sensation than intradermal hydrocortisone, hyaluronidase and antioxidants [106]. The results by Lee et al. showed that glabridin inhibited the myofibroblast features of fBMFs through the TGF-B/Smad2 signaling pathway and that it also prevented arecoline increased fBMFs activity [80]. So, it can be used as a natural antifibrotic compound to treat OSF. Hsieh et al. found that EGCG inhibited arecoline-induced activation of TGF-β1, mitochondrial ROS and subsequent synthesis of CTGF and Egr-1 in BMFs in a dose-dependent manner [23,107]. Studies by Nerkar Rajbhoj et al. and Al-Maweri et al. have shown that curcumin is effective in improving OSF symptoms without any side effects [98,108]. Several studies have shown that the administration of Aloe vera to patients with OSF can reduce the clinical symptoms of patients and its clinical efficacy is just like that of intra-lesion injection of hydrocortisone and hyaluronidase with antioxidant supplementation [98,106,109,110]. Honokiol is a polyphenolic component derived from Magnolia officinalis. Chen et al. showed that the expression of the TGF-β/Smad2 pathway was downregulated and the expression of α-SMA and type I collagen was downregulated when treated with honokiol, and honokiol could inhibit arecoline-induced fBMFS activity [111,112]. It could prevent the oral squamous cell epithelium from turning into cancer [113] and is a promising compound.

The treatment methods and principles of OSF are listed in Table 1.

## 8. Summary

The above studies on the mechanism of OSF and malignant transformation into OSCC make the key molecules and abnormal RNAs clear, which are of great significance for the occurrence, development, detection, diagnosis, monitoring and prognosis of the disease. The important molecules and processes involved in OSF pathogenesis and malignant transformation to OSCC are shown in Figure 3. The roles of miRNAs and the targets mentioned above for OSF and OSCC are listed in Table 2. The pathways and roles of the mentioned lncRNAs and circRNAs in OSF and OSCC are listed in Table 3.

Therefore, targeted blocking of these key molecules is the trend and key for disease prevention and treatment in the future. The application of natural compounds has also shown its therapeutic potential, so it is necessary to conduct further research on natural compounds in the hope that they will become first-line drugs for OSF treatment in the future.

## Figures and Tables

**Figure 1 ijms-24-04992-f001:**
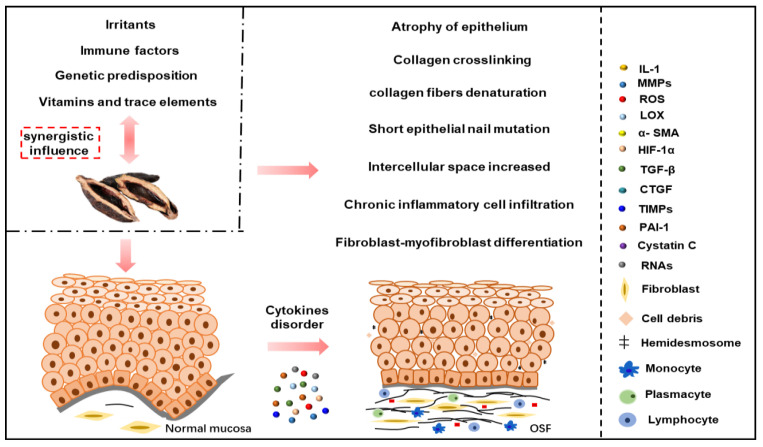
A multi-factor pathogenic model of OSF with BQ as the main cause. Firstly, BQ components can play a pathogenic role in conjunction with tobacco and alcohol, vitamin and trace element abnormalities, immune factors and genetic susceptibility, and directly stimulate keratinocytes and fibroblasts and cause ulcers, inflammation, oxidative stress, cell senescence, cytotoxicity, epithelial atrophy, etc. Secondly, the above factors can also increase collagen synthesis, collagen cross-linking, reduce collagen degradation, fibroblast–myofibroblast differentiation and activation by regulating TGF-β, ROS, LOX, MMPs, TIMPs, PAI-1, Cystatin C, HIF-1α, ncRNAs and other signal molecules. Fibrosis leads to vascular occlusion, reduced blood supply and hypoxia, aggravating the vicious cycle of OSF and making fibrosis even more irreversible.

**Figure 2 ijms-24-04992-f002:**
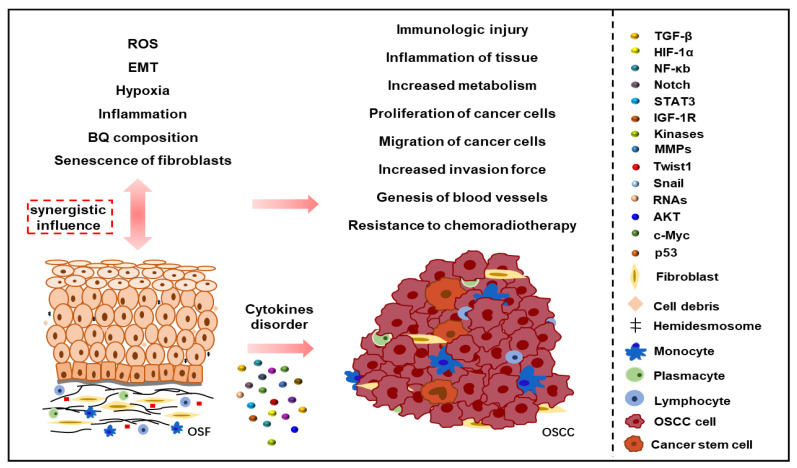
Model of malignant transformation based on OSF. BQ and its carcinogenic components (areca alkaloids, hydrated lime, nitrosamines, polyphenols, etc.), the senescence of fibroblasts, EMT, the continuous production of ROS, the aggravation of hypoxia, the continuous inflammatory environment and the aggravation of fibrosis will lead to malignant transformation through the regulation of TGF-β, HIF-1α, NF-κB, Notch, STAT3, IGF-1R and other pathways and cytokines on the basis of OSF. This will lead to the generation of cancer stem cells, affect cell growth, migration, invasion, metabolism, and even resistance to chemotherapy and radiotherapy, and promote the occurrence and progression of OSCC on the basis of OSF.

**Figure 3 ijms-24-04992-f003:**
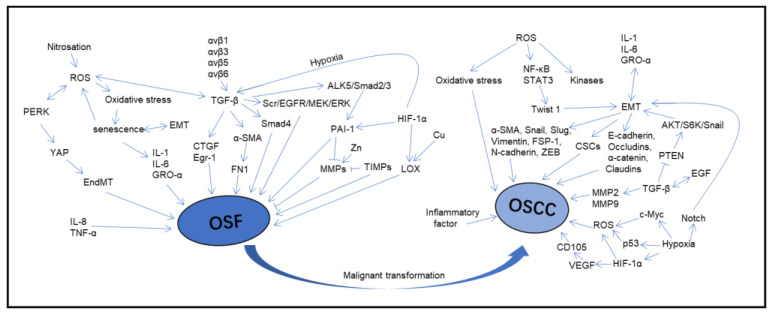
The important molecules and processes involved in OSF pathogenesis and malignant transformation into OSCC.

**Table 1 ijms-24-04992-t001:** Treatment methods and principles of OSF.

Methods	Principles	Name	Ref
Oral health education	Prevention is better than cure		
Removal of pathogenic factors	Intervention		
Chemical drug therapy	Glucocorticoids: inhibit the production of inflammatory factors, promote the apoptosis of inflammatory cells, are anti-inflammatory and inhibit fibrosis	Triamcinolone (mixture of triamcinolone 30 mg + lidocaine 20 mg, local injection once a week, eight times as a course of treatment) Dexamethasone (mixture of dexamethasone 4 mg + hyaluronidase 1500 IU + chymotrypsin 5000 IU, two injections per week for six to eight weeks)	[92,93,99,114]
	Antifibrotic drugs and proteolytic enzymes: proteolytic, hydrolyzes hyaluronic acid and reduces collagen formation	Hyaluronidase (hydrocortisone acetate 25 mg/mL + hyaluronidase 1500 IU, local injection once a week for six weeks)	[93,98]
	Peripheral vasodilator: improves microcirculation and hemorheology	Pentoxifylline (400 mg, three times daily for three to four months) Isoxuprine (10 mg, three times daily for four to six weeks)	[94,95,114]
	Oxidants and nutrients: antioxidant and deactivation of free radicals	Lycopene (8 mg, once a day for half a year) Vitamin A (50,000 IU, once a day for half a year) Vitamin B, C, D Vitamin E (400 mg, once a day for half a year)	[96,97,114]
Oral physiotherapy exercises	Physical therapy	Mouth opening training	[98,99]
Surgical treatment	The fibrous bands were excised and different flaps were implanted	Fiber strip excision	[92]
Hyperbaric oxygen therapy	Increase blood oxygen content, improve local ischemia and hypoxia, promote neovascularization and collateral circulation in the damaged area	Hyperbaric oxygen	[103]
Laser therapy	Ease mouth opening difficulty, burning sensation and increase cheek flexibility	Laser loosens the fiber band	[104]
Natural compounds	Anti-ibrosis, anti-inflammation and anti-tumor	Arctigenin	[12]
	Invigorate the circulation of blood	Salvia miltiorrhiza (mixture of salvia miltiorrhiza 300 mg + lidocaine 20 mg, local injection once a week, eight times as a course of treatment)	[99,105]
	Anti-inflammatory, antioxidant, anti-ulcer and antifibrotic	Black turmeric (black turmeric 0.5 mg + Aloe vera gel 0.5 mg, three times a day for three months)	[106]
	Anti-inflammatory, antioxidant	Glabridin	[80]
	Antioxidant, inhibit collagen synthesis	Epigallocatechin gallate	[23,107]
	Modulates the inflammatory response, decreases collagen generation and induces apoptosis	Curcumin (300 mg, once daily, six to eight months)	[98,108,114]
	Antioxidant, anti-inflammatory, immunomodulatory and anti-tumor	Aloe vera (5 mg of Aloe vera gel, smearing, six weeks)	[98,106,109,110]
	Anti-inflammatory, antifibrotic and anti-cancer	Honokiol	[111,112]

**Table 2 ijms-24-04992-t002:** Targets and roles of miRNAs in OSF and OSCC.

miRNA	Change	Targets	Roles	Ref
miR-29c	Downregulation	TPM1 binds directly to miR-29c.	Downregulation of miR-29c affects TPM1 and myofibroblast activity.	[28]
miR-204	Downregulation	Predicted target: MMP9, PLAUR, SERPINE1, SNAI2, COL5A3.	MiR-204 is considered to be a tumor suppressor. Restoring its expression can significantly inhibit cell proliferation, migration, invasion and tumor formation, and significantly increase the rate of cell apoptosis.	[41]
miR-509-5p	Downregulation	Predicted target: BMPR2, CDH6, HAS3, PARD6B, TIMP3, THBS1, THBS2.	Inhibited cell proliferation and migration, promote apoptosis in OSF.	[42]
miR-610	Downregulation	Predicted target: CDH1, DSC2, KRAS, MMP19, MAPK1, TIMP3, MMP24.	It could cause collagen defect of OSF.	[42]
miR-760	Downregulation	Histone mRNA binds directly to miR-760. Predicted target: CDH4, COX10, IL6, IL6R, IGF1R, TIMP2, TGM2.	MiR-760 can be used as a predictor of precancerous lesions. KEGG pathway analysis has demonstrated the role of the hedgehog signaling pathway and ubiquitin-mediated proteolysis.	[42,43]
miR-200C	Downregulation	ZEB1 binds directly to miR-200c.	Overexpression of miR-200c inhibits the expression of ZEB1 and α-SMA, and it is a key factor in tumorigenesis and cancer metastasis.	[46]
miR-203	Downregulation	COL4A4 and miR-203 were negatively correlated.	It could inhibit fibrosis and epithelial–mesenchymal transition.	[48]
miR-21	Upregulation	PDCD4 binds directly to miR-21.	MiR-21 inhibition of PDCD4 can regulate myofibroblast activation of BMFs and increase the invasiveness of oral cancer.	[39,82]
miR-31	Upregulation	Predicted target: DMD, CXCL12, WASF3.	High expression of mir31 can affect the immortalization or transformation of oral keratinocytes, damage DNA repair genes, and cause genomic instability and EMT transition.	[41]
miR-623	Upregulation	Predicted target: MAPK1, MAPK11, MAPK4, MMP1, MMP8, TIMP2, IL10.	It may be a profibrotic miRNA. The KEGG pathway included gap junction, TGF-β signaling pathway, insulin signaling pathway, chemokine signaling pathway, and cytokine–cytokine receptor interaction.	[42]
miR-455	Upregulation	BMP7 was a negative modulator of fibrosis. Predicted target: BMPR, DSC1, MAPK14, MAPK11, IGF1, TIMP2, TGM3, BMP7.	It may have a promoting role in the OSF.	[42,44]
miR-10b	Upregulation	Regulated by Twist.	MiR-10b regulates the activation of myofibroblasts and the expression of α-SMA.	[45]
miR-1246	Upregulation	Type I collagen and miR-1246 were positively correlated.	The activity of collagen gel contraction and the migration ability of fibroblasts was inhibited by inhibiting miR-1246.	[47]

**Table 3 ijms-24-04992-t003:** Pathways and roles of lncRNAs and circRNAs in OSF and OSCC.

RNA	Change	Pathways	Roles	Ref
LINC00974	Upregulation	TGF-β/Smad pathway	It promotes the development of oral fibrosis.	[12]
LINC00084	Upregulation	LINC00084/miR-204/ZEB1 axis	LINC00084 regulates myofibroblast activation through miR-204-ZEB1.	[49]
lncRNA H19	Upregulation	IncRNAH19/miR-29b/COL1A1 axis	Chewing betel nut can upregulate the expression of H19 in BMFs by activating TGF-β1. lncRNA H19 acts as a molecular sponge for miR29b and causes oral fibrosis by interfering with the binding of miR-29b to collagen type I.	[50]
circEPSTI1	Upregulation	circEPSTI1/miR-942-5p/LTBP2 axis and PI3K/Akt/mTOR pathway	CircEPSTI1/miR-942-5p/LTBP2 axis activates EMT and PI3K/Akt/mTOR signaling pathways to promote OSCC proliferation, migration and invasion.	[83]
lncRNA MEG3	Downregulation	IncRNA MEG3/miR-421	IncRNA MEG3 affects the characteristics of cancer stem cells through the molecular sponge miR-421.	[84]

## Data Availability

Not applicable.

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
