# Peer review of "Oral Submucous Fibrosis: Etiological Mechanism, Malignant Transformation, Therapeutic Approaches and Targets"

_ijms, 2023, doi:10.3390/ijms24054992_

Round 1
Reviewer 1 Report
The paper is an excellent review on the association between chewing betel and oral submucous fibrosis, and extensively discusses the diagnostic, therapeutic and molecular aspects of OSF as well. The article also deals with the possibility and mechanisms of malignant transformation of the lesions. The topic is important, and the paper well synthesize the theoretical knowledge with practical/clinical issues; it is worth for publication.
Minor shortcomings, however, still can be identified. In contrast to the title, the paper focuses on the role of betel in the development of OSF. The other possible risk factors are briefly listed under 3.2-3.6, but the authors do not describe the possible mechanisms/pathways how these factors could lead to OSF. The paper is undoubtedly betel-oriented - there is no problem with this, since the major risk factor is betel use, but, based on the title the reader expects more information of other mechanisms as well. The simplest solution would be to change the title so that it better reflects the content of the article.
Another minor issue is the section "Epidemiology" which is rather minimalist. This paragraph should be extended with incidence data to characterize the occurrence of the disease, and to write some words about time-trends in the incidence.
Finally, the OSF and OSCC abbreviations sometimes contain 0 (zero) instead of O (capital o) which is disturbing.
Author Response
Thank you for your constructive comments, which we think are very necessary, so we have revised every point. The following is the detailed explanation.
Question 1. The other possible risk factors are briefly listed under 3.2-3.6, but the authors do not describe the possible mechanisms/pathways how these factors could lead to OSF.
Answer: We have combined and supplemented the risk factors of OSF in section 3 and the mechanism in section 4. The revised section 3 (risk factors and pathogenesis of OSF) has detailed introduction under each subtitle.
Question 2. Another minor issue is the section "Epidemiology" which is rather minimalist.
Answer: We have supplemented the epidemiology of OSF in section 2 with respect to the incidence and prevalence trends.
Question 3. Finally, the OSF and OSCC abbreviations sometimes contain 0 instead of O.
Answer: We have checked the spelling of the words in the review several times and corrected the problems.
Thank you very much for reviewing our article. If there is anything that needs to be improved, please advise again.
Reviewer 2 Report
Manuscript Title: Oral Submucous Fibrosis: Etiological Mechanism, Malignant Transformation and Therapeutic Target
Date: 20.11.2022
Thank you for asking me to assess the above-titled manuscript. Comments for the Authors GENERALThis manuscript presents a review of OSF, and attempt to summarize the key molecules in the pathogenic, malignant mechanism and management of OSF.
Here are my observations on this study/paper:
Abstract
Entirely appropriate.
Sections 1-10
· This is narrative review with a lot of details especially pathogenic mechanism and malignant transformation, but the study lack essential details in some sections.
· There are a number of up to date references which should be included and omit the outdated one.
· OSF not (0SF) and OSCC not 0SCC, please correct.
· Expand sections 3.2, 3.3, 3.4, 3.5 and 3.6
· Expand section 8.3. Drug therapy with emphasis on mode of delivery, doses, effectiveness
· Section 8.4, Auxiliary mouth opening training, expand this section with focus in its effectiveness
· Section 8.5, more details are needed and highlight the drawback of surgical treatment
· Please highlight the limitations of the narrative review as compared to systematic review.
· Consider adding short but detailed summary of study, and shorten the conclusion section.
References
Some of the references do not conform to the journal style.
Recommendation
· The present study has several limitations. First, the lack of in-depth details in some sections as mentioned above, second, this is narrative review and the limitation of this study design should be highlighted at the start of the study
· It may be helpful for the reader if you summarize the findings at the end of the study.
· Number of typos were noted, perform carful check of the document.
Author Response
Thank you for your constructive comments, which we think are very necessary, so we have revised every point. The following is the detailed explanation
Question 1. There are a number of up to date references which should be included and omit the outdated one.
Answer: We cited new references and replaced some of the outdated one. The new references in the revised version are 6, 7, 14, 15, 21, 29, 99,100, 101, 102, 114.
Question 2. OSF not (0SF) and OSCC not 0SCC, please correct.
Answer: We have checked the spelling of the words in the review several times and corrected the problems.
Question 3. Expand sections 3.2, 3.3, 3.4, 3.5 and 3.6.
Answer: We have combined and supplemented the risk factors of OSF in section 3 and the mechanism in section 4. The revised section 3 (risk factors and pathogenesis of OSF) has detailed introduction under each subtitle.
Question 4. Expand section 8.3. Drug therapy with emphasis on mode of delivery, doses, effectiveness.
Answer: We supplemented Drug therapy according to existing clinical studies, and the supplementary results were summarized in Table 1.
Question 5. Section 8.4, Auxiliary mouth opening training, expand this section with focus in its effectiveness
Answer: In the revised section 7.4 (Auxiliary mouth opening training), we have added related clinical studies on mouth opening training.
Question 6. Section 8.5, more details are needed and highlight the drawback of surgical treatment
Answer: We have added the drawback of surgical treatment in the revised section 7.5 (Surgical treatment).
Question 7. Please highlight the limitations of the narrative review as compared to systematic review.
Answer: A systematic review is a study that uses explicit methods, queries, selects and rigorously evaluates relevant studies, extracts data from them and combines them with appropriate statistical methods to present comprehensive conclusions. Finally hope to solve a problem to provide a theoretical basis. The important thing about systematic reviews is evaluation. Compared with traditional reviews, systematic reviews usually focus on a specific clinical problem, strive for comprehensive literature sources, and have clear retrieval steps. There are strict inclusion and exclusion criteria for literature selection, qualitative and quantitative evaluation of data synthesis, and strict evaluation methods. Traditional review is an academic paper in which the author collects a large amount of relevant literature and materials for a certain field, profession or research topic according to the specific purpose or interest. Based on extensive reading and understanding, the author adopts a qualitative method to make a comprehensive introduction and elaboration of the research status quo, latest progress, academic opinions or suggestions in the field through comprehensive analysis, summary, sorting and refining. Traditional review can help readers understand the general situation and development direction of a certain topic in a short time, but it is subjective and lacks objective methods. In addition, the data collected by traditional reviews is not very comprehensive.
Question 8. Consider adding short but detailed summary of study, and shorten the conclusion section.
Answer: We combine section 9 (Prevention and treatment orientation) and section 10 (Conclusion) in the original version into section 8 (Summary) in the modified version. Figure3 and table2, 3 are used to make a brief summary of the above.
Question 9. Some of the references do not conform to the journal style.
Answer: A few references were deleted and replaced.
Thank you very much for reviewing our article. If there is anything that needs to be improved, please advise again.
Reviewer 3 Report
This article aims to provide a review of OSF on the pathogenic factors, mechanisms and potential therapeutic targets. Although it is an interesting topic, the quality of this review in its current form is not up to the publication standard of this journal at all. There are several major concerns about the overall construction and focus of this article as well as the quality of the figures that require substantial revisions and re-written by the authors. The general impression of this reviewer is that this article is not well synthesised and written, with some parts being too lengthy and out of focus. It lacks balance and some parts, such as disease management, are not quite relevant to the topic of this review.
Construction and quality issues:
There are some overlaps in the context of the mechanisms of OSF and OSCC in Sections 4 and 6. Can these two sections merge with an altered main subheading and be further separated by subheadings, uch as BQ, TGF-beta, etc.?
‘Section 7. Diagnosis’ should be moved to the position after Section 3.
Given the title of this review, Sections 7 & 8 contain too much information and need to be shortened. ‘Section 8. Disease management’ is too lengthy and can be considered to be published elsewhere in a clinical-related journal. Alternatively, the authors should change the title of the article.
non-coding RNAs (ncRNAs), Long Non-coding RNAs (lncRNAs), Circular RNAs (circRNAs) should be described as part of the mechanisms.
Table 2. needs to be re-arranged with all the downregulated and upregulated miRNA separately and the text related to it needs to be improved.
Section 3: suggest removing the subtitle. Otherwise, each subheading needs to be expanded with additional descriptions.
Section 2: lack the incidence data
Line 93: ‘the phenotype of cells were increased’ what does this mean?
Line 173: there is a lack of description of HIF-1α beforehand
Line 291: what are the ‘L fibroblasts’?
The quality of the figures is poor and the images used in the figure do not mean much.
Punctuation and format issues:
A thorough proofreading is needed as there are errors in numerous places and punctuation and space issues, etc. The font size varies in some sentences, e.g. Line 275-281
In many places, OSF is expressed as ‘0SF’ and in places as ‘osf’. The same applies to AVB6, TGFb1, etc.
Abbreviation issue:
When the abbreviation is introduced for the full term at its first mention, use the abbreviation from then on. However, there are places with full text reappearing, e.g. BQ (used in the Figure 1 legend but the full term is used in the Figure 2 legend), OPMDs, etc.
Line 51: what does OSMF stand for in its first appearance?
ROS was first introduced at Line 130 but the full term shown at Line 240
Add a section of abbreviations at the end of the article
Author Response
Thank you for your constructive comments, which we think are very necessary. The following is a detailed explanation of the changes for each point.
Question 1. There are some overlaps in the context of the mechanisms of OSF and OSCC in Sections 4 and 6. Can these two sections merge with an altered main subheading and be further separated by subheadings, uch as BQ, TGF-beta, etc.?
Answer: We have combined and supplemented the risk factors of OSF in section 3 and the mechanism in section 4. The revised section 3 (risk factors and pathogenesis of OSF) has detailed introduction under each subtitle. It is true that there is some overlap between the mechanisms of OSF and the malignant transformation into OSCC, but not all OSF will progress to OSCC. OSCC also has many other pathogenic factors and mechanisms, so we want to separate these two parts and describe them gradually.
Question 2. ‘Section 7. Diagnosis’ should be moved to the position after Section 3.
Answer: There are some descriptions of malignant transformation of OSF in the diagnosis, so it is considered more appropriate to put diagnosis before treatment.
Question 3. Given the title of this review, Sections 7 & 8 contain too much information and need to be shortened. ‘Section 8. Disease management’ is too lengthy and can be considered to be published elsewhere in a clinical-related journal. Alternatively, the authors should change the title of the article.
Answer: We changed the title to “Oral Submucous Fibrosis: Etiological Mechanism, Malignant Transformation, Therapeutic Approaches and Targets “, and in the title we also emphasized the therapeutic approaches.
Question 4. non-coding RNAs (ncRNAs), Long Non-coding RNAs (lncRNAs), Circular RNAs (circRNAs) should be described as part of the mechanisms.
Answer: We described ncRNAs in the new version of 'section 3 Risk factors and pathogenesis of OSF'.
Question 5. Table 2. needs to be re-arranged with all the downregulated and upregulated miRNA separately and the text related to it needs to be improved.
Answer: In the new version, we have reworked Table 2 and adjusted the position of its associated text.
Question 6. Section 3: suggest removing the subtitle. Otherwise, each subheading needs to be expanded with additional descriptions.
Answer: We have combined and supplemented the risk factors of OSF in section 3 and the mechanism in section 4. The revised 'section 3 risk factors and pathogenesis of OSF' has detailed introduction under each subtitle.
Question 7. Section 2: lack the incidence data.
Answer: We have supplemented the epidemiology of OSF in section 2 with respect to the incidence and prevalence trends.
Question 8. Line 93: ‘the phenotype of cells were increased’ what does this mean?
Answer: I'm sorry, this problem is caused by semantic ambiguity. The correct meaning is, 'the proliferation of BMFs increased and the phenotype of cells changed'. This sentence is in section 3.5 of the modified version, line 162, and we have corrected this mistake in the new version.
Question 9. Line 173: there is a lack of description of HIF-1α beforehand
Answer: In the new version of section 3.5, the related description of HIF-1α is added.
Question 10. Line 291: what are the ‘L fibroblasts’?
Answer: Sorry for not checking carefully enough, 'L' was wrongly added, we have corrected this mistake in the new version.
Question 11. The quality of the figures is poor and the images used in the figure do not mean much.
Answer: In the new version, we have improved the figure and replaced the old images.
Question 12. Punctuation and format issues.
Answer: We have repeatedly checked the punctuation, formatting and spelling of the words in the review and corrected the mistakes.
Question 13. Abbreviation issue.
Answer: We checked and corrected the nonstandard term abbreviation in the review several times and introduced this abbreviation at the first mention of the full term. A section of abbreviation was added at the end of the article.
Thank you very much for reviewing our article. If there is anything that needs to be improved, please advise again.